# Phylogenetic divergence of cell biological features

Michael Lynch*

Center for Mechanisms of Evolution, Biodesign Institute, Arizona State University, Tempe, Arizona

**Abstract** Most cellular features have a range of states, but understanding the mechanisms responsible for interspecific divergence is a challenge for evolutionary cell biology. Models are developed for the distribution of mean phenotypes likely to evolve under the joint forces of mutation and genetic drift in the face of constant selection pressures. Mean phenotypes will deviate from optimal states to a degree depending on the effective population size, potentially leading to substantial divergence in the absence of diversifying selection. The steady-state distribution for the mean can even be bimodal, with one domain being largely driven by selection and the other by mutation pressure, leading to the illusion of phenotypic shifts being induced by movement among alternative adaptive domains. These results raise questions as to whether lineage-specific selective pressures are necessary to account for interspecific divergence, providing a possible platform for the establishment of null models for the evolution of cell-biological traits.
DOI: https://doi.org/10.7554/eLife.34820.001

## Introduction

As with nearly all biological traits, most cellular features vary among individuals within populations in a nearly continuous fashion, owing to genetic differences among individuals and the myriad of stochastic factors experienced by all organisms (ranging from intrinsic cellular noise to external environmental forces; *Lynch and Walsh, 1998*). This is true, for example, for catalytic rates, rates of gene expression and intracellular transport, numbers and sizes of organelles, etc. Ultimately, some fraction of within-species genetic variation is transformed into among-species divergence as alternative alleles arise by mutation and in some cases proceed to fixation (*Wright, 1969*; *Walsh and Lynch, 2018*). The magnitude of such divergence is dictated by three major evolutionary factors: the pattern of selection (the phenotypic fitness function), which imposes a directional and/or stabilizing force on the mean phenotype; the rate of origin and distribution of mutational effects, which define the raw materials upon which natural selection operates; and the power of random genetic drift, which imposes noise on the selective process.

Although considerable effort has been devoted to understanding the divergence of mean phenotypes among lineages (*Walsh and Lynch, 2018*), most of this work is focused on the evolution of morphological phenotypes in response to external pressures, which can vary greatly depending on the ecological setting. In contrast, owing to homeostatic effects, the internal environment of cells remains largely constant over long time scales and broad geographic locations, raising the possibility of establishing general evolutionary principles that transcend the imposition of transient ecological changes. (The same might be true for the internal organs of multicellular species).

The goal here is to derive general expressions for the divergence of mean phenotypes among species under scenarios that are likely to hold for a wide variety of cellular traits. The specific focus will be on the magnitude of divergence expected among lineages in the face of identical evolutionary forces, as this helps clarify the degree to which phenotypic diversification can proceed in the absence of lineage-specific selection pressures. Such a perspective is essential to establishing the

*For correspondence:
mlynch11@asu.edu

Competing interests: The author declares that no competing interests exist.

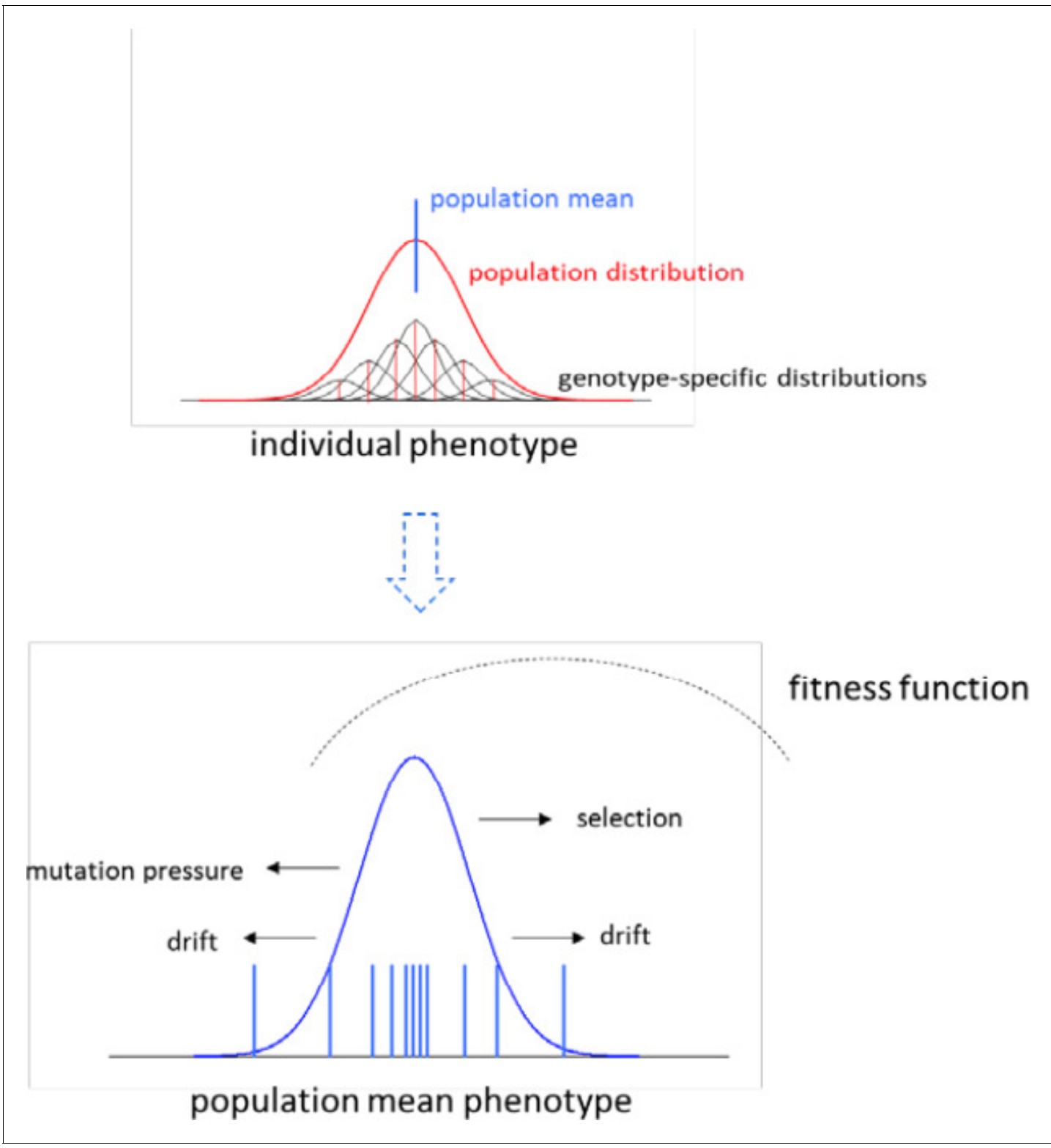

**Figure 1.** An idealized overview of the model for the evolution of the distribution of mean phenotypes, given here for a trait under stabilizing selection. The upper panel denotes a hypothetical phenotype distribution at a single point in time. The population consists of multiple genotypes, each having an expected genotypic value (red) but a range of phenotypes (black distributions) resulting from variance in residual deviations (environmental effects and nonadditive genetic factors). The phenotype distribution for the entire population (red) is the sum of these genotype-specific curves, and has a mean denoted by the blue line. The exact location of this overall distribution can wander over time, owing to the joint forces of selection, mutation, and random genetic drift. The lower panel gives the overall distribution of population means over a long evolutionary time span, with 11 locations at

*Figure 1 continued on next page*

*Figure 1 continued*
specific points of time being denoted by the short vertical lines. Persistent mutational bias towards smaller phenotypes prevents the overall distribution of means from coinciding with the fitness-function optimum, and random genetic drift causes a dispersion of means around the overall average value.
DOI: https://doi.org/10.7554/eLife.34820.003

degree to which adaptive explanations need to be sought to explain patterns of variation among populations.

The general approach will draw from well-established constructs employed in the field of quantitative genetics (the study of continuously distributed traits with a multifactorial genetic basis; *Lynch and Walsh, 1998*; *Walsh and Lynch, 2018*). The traditional focus of this field has been on complex traits in multicellular species, but these same methods can be profitably applied to intracellular morphological and molecular features, such as those involved in the cytoskeleton, gene expression, binding energy, and metabolic rates (*Nourmohammad et al., 2013*; *Farhadifar et al., 2015*; *Phillips and Bowerman, 2015*). Indeed, although most work in phenotypic evolution proceeds as though cellular details are irrelevant, the models employed may be equally if not more relevant to cell-biological traits, owing to their potentially less temporally variable fitness effects.

## Theory

### The distribution of mean phenotypes

All genetically encoded traits are subject to the recurrent forces of mutation and random genetic drift, and potentially to selection. Selection favors some genotypes over others, while mutation modifies existing genotypes independent of the selective process, and random genetic drift causes stochastic variation in gene transmission across generations. Owing to this latter factor, even if the forces of selection and mutation remain constant, the population mean phenotype of a trait will wander within a certain range over evolutionary time, with the frequency of occurrence of alternative mean phenotypes depending on patterns and strengths of selective and mutational effects (*Figure 1*).

The focus of this study, the stationary distribution of mean phenotypes, can be viewed as a summary distribution of: (1) phenotypic means across a large number of replicate populations exposed to identical conditions for a very long period; or (2) a historical survey of mean phenotypes in a single population over a long time period, again under constant environmental and population-genetic conditions. Among many other applications, such an approach has long been exploited in attempts to understand the steady-state distribution of allele frequencies expected under a constant regime of selection, mutation, and random genetic drift (e.g. *Wright, 1969*). From an empirical perspective, this steady-state view of evolution implicitly assumes that enough time has elapsed between observed taxa that the dynamics of the evolutionary process are of negligible significance (which would not be the case for closely related species).

The approach taken here relies on the Kolmogorov forward equation for a diffusion process (Appendix 1, *Walsh and Lynch, 2018*), the assumption being that the trait of interest is continuously distributed, with $z$ denoting the phenotypic value of an individual. The population mean, $\bar{z}$, moves in arbitrarily small increments each generation via the deterministic forces of selection and mutation and the stochastic process of drift. Under most reasonable biological conditions, independent of the starting conditions, a stationary distribution of mean phenotypes (among hypothetical replicate populations) is eventually converged upon, at which point there is an exact balance between opposing forces. The probability that a population's mean phenotype will reside at any particular point is defined by this distribution, which has the general form

$$\Phi(\bar{z}) = C \cdot \exp\left( 2 \int^{\bar{z}} [M(x)/V(x)]dx \right),\tag{1a}$$

where $M(x)$ defines the rate of directional change (resulting from selection and/or mutation) for a population with mean phenotype $x$, and $V(x)$ is the variance in change (resulting from drift). $C$ is the normalization constant (containing only terms that are independent of $\bar{z}$) that ensures that the entire probability density sums to 1.0.

For a quantitative trait, the directional term can be subdivided into independent selection and mutation components, $M_s(x)$ and $M_m(x)$, both of which will be discussed in detail below. Under the assumption of negligible genotype $\times$ environment interaction and epistasis, the variance of the change in means, which results from the sampling of heritable genotypic values of individuals, is equal to the underlying additive genetic variance for the trait, $\sigma_A^2$, divided by the effective population size, $N_e$, in the case of haploidy (assumed here; and $2N_e$ in the case of diploidy). The latter is typically far below the number of reproductive individuals in the population, and defined by various demographic features and interference imposed by chromosomal linkage, with values ranging between $\sim 10^5$ for multicellular eukaryotes to $\sim 10^9$ for bacteria (*Charlesworth, 2009*; *Lynch et al., 2016*; *Walsh and Lynch, 2018*).

Individual phenotypes are comprised of the sum of a heritable additive genetic component ($A$) and a nonheritable residual deviation ($e$, which includes environmental and nonadditive genetic effects), such that $z = A + e$, with the within-population phenotypic variance being partitioned as $\sigma_z^2 = \sigma_A^2 + \sigma_e^2$. For cellular features, a large fraction of $\sigma_e^2$ may be a consequence of stochastic gene expression, imprecise placement of cell-division septa, etc. Assuming that both $\sigma_A^2$ and $N_e$ remain constant, which is the model adhered to here, *Equation (1a)* can be rewritten as

$$\Phi(\bar{z}) = C \cdot \exp\left( (2N_e/\sigma_A^2) \int^{\bar{z}} [M_s(x) + M_m(x)] dx \right),$$ (1b)

showing that the stationary distribution of mean phenotypes (conditional on a particular level of genetic variance, a point that will be returned to below) is proportional to the product of the distributions expected under selection alone and under mutation alone. With extremely weak selection, $M_s(x)$ would be essentially a flat function, with the overall distribution reflecting the biases due to mutation alone. Conversely, with a flat mutation function, an unlikely scenario, the distribution will follow that expected under selection alone.

## The process of selection

The influence of selection on the mean phenotype (the response to selection) is embodied in the breeder's equation,

$$\begin{aligned} M_s(\bar{z}) \ &= \bar{z}(t+1) - \bar{z}(t) \\ &= h^2 [\bar{z}_s(t) - \bar{z}(t)], \end{aligned}$$ (2)

a general statement about the connection between directional selection within generations and the transmission of such change across generations (*Walsh and Lynch, 2018*). Here, $\bar{z}(t)$ and $\bar{z}_s(t)$ denote the mean phenotypes before and after selection in generation $t$, the difference being the selection differential. The heritability of the trait, $h^2 = \sigma_A^2/\sigma_z^2$, which equals the proportion of the total phenotypic variance, $\sigma_z^2$, associated with additive genetic variation, $\sigma_A^2$, constitutes the fraction of the within-generation change in the mean transmitted to the next generation.

Critical to everything that follows, the selection differential can be described in terms of the within-population phenotype distribution, $p(z,t)$, and the function relating individual fitness to phenotype, $W(z)$. The mean fitness in generation $t$ is

$$\overline{W} = \int p(z,t) \cdot W(z) \cdot dz.$$ (3)

The mean phenotype after selection (but before inheritance) is then obtained by weighting the preselection phenotypes by their relative fitnesses,

$$\bar{z}_s(t) = \frac{1}{\overline{W}} \int z \cdot p(z,t) \cdot W(z) \cdot dz.$$ (4)

We will make use of the fact that most quantitative traits have an approximately normal phenotype distribution on some scale of measurement, which follows from the central limit theorem (*Lynch and Walsh, 1998*). The distribution of individual measures is therefore described completely by the phenotypic mean and variance,

$$p(z,t) = \frac{1}{\sqrt{2\pi\sigma_{\bar{z}}^2}} \cdot \exp\left(\frac{-[z - \bar{z}(t)]^2}{2\sigma_{\bar{z}}^2}\right). \tag{5}$$

Substituting *Equation (5)* into (3) and differentiating, the change in mean fitness with respect to mean phenotype is

$$\begin{aligned}
\frac{\partial \overline{W}}{\partial \bar{z}(t)} &= \int \frac{\partial p(z,t)}{\partial \bar{z}(t)} \cdot W(z) \cdot dz \\
&= \frac{1}{\sigma_{\bar{z}}^2} \int [z - \bar{z}(t)] \cdot p(z,t) \cdot W(z) \cdot dz
\end{aligned} \tag{6}$$

(*Lande, 1976*). From *Equation (4)*, the first term to the right of the integral is equal to $\bar{z}_s(t) \cdot \overline{W}$, and the second term is $\bar{z}(t) \cdot \overline{W}$. This provides a direct link to *Equation (2)*, which upon rearrangement becomes

$$M_s(\bar{z}) = \sigma_A^2 \cdot \frac{\partial \overline{W}}{\overline{W} \cdot \partial \bar{z}(t)}. \tag{7}$$

This expression states that, provided the phenotype distribution is normal, the change in mean phenotype caused by selection is equal to the product of the genetic variance for the trait and the gradient in the logarithm of mean fitness with respect to mean phenotype. Evolution by natural selection comes to a standstill when there is no genetic variance for the trait or the phenotypic mean resides at a point where the slope of the function of mean fitness with respect to mean phenotype is zero. To endow this expression with practical utility, specific expressions for the fitness function, $W(z)$, will be considered below.

## The process of mutation

Most attempts to consider the long-term evolutionary features of quantitative traits have assumed one of two mutation models: (1) a distribution of mutational effects always having a mean equal to zero and a constant variance, independent of the starting genotype (*Kimura, 1965*; *Lande, 1975*; *Lynch and Hill, 1986*); or (2) a rate of appearance of each type of mutant allele being independent of the ancestral type (*Cockerham, 1984*; *Turelli, 1984*). Under the first scenario, mutation has no directional effect on the mean phenotype, and there are no bounds on the possible mutational effects or the physical limits to which the trait can evolve. Under the second scenario, there is a physical limit to phenotypic divergence, and because the directional effect of mutations depends on the current location, more extreme alleles generate mutations with effects biased back toward the center of the distribution.

Neither of these mutational schemes captures the features of a wide variety of cell biological traits, which often have finite numbers of possible states and state-dependent spectra of mutational effects. A few examples will suffice to make this point. Protein-protein interactions (e.g. the interfaces between dimeric molecules) typically depend on no more than a few dozen amino-acid sites. The same is true for intramolecular interactions such as the constellation of backbone residues that assemble during protein folding. In both cases, the underlying residues operate in an approximately binary manner, for example, hydrophobic vs. hydrophilic, or hydrogen-bonding vs. non-hydrogen bonding. Likewise, the catalytic sites of enzymes often consist of a small-to-moderate numbers of residues that either facilitate or inhibit catalytic rates, and the sizes of intracellular organelles and cytoskeletal components are constrained by cell size. Many other examples could be cited, including those involved in RNA-RNA and DNA-protein interactions.

The approximate structure of a mutation function with a bounded range can be arrived at by considering a trait determined by $n$ binary factors (or sites), each with state b having effect 0, and state B having effect $m$. For a trait with an additive genetic basis, the mean phenotype in a haploid population can then be represented as

$$\bar{z} = z_0 + nm\bar{q}, \tag{8}$$

where $z_0$ is an arbitrary baseline value for the trait, and $\bar{q}$ is the mean frequency of B-type alleles averaged over all $n$ factors in the population (*Lynch and Walsh, 1998*).

Letting $u$ be the mutation rate from B to b alleles, and $v$ be the reciprocal rate, the per-generation change in the mean phenotype resulting from mutation is

$$M_m(\bar{z}) = nm[v(1 - \bar{q}) - u\bar{q}]. \tag{9}$$

With $q = v/(u + v)$ being the equilibrium frequency of B alleles under mutation pressure alone, and $\theta_m = z_0 + nmq$ being the expected mean phenotype under neutrality, *Equation (9)* further reduces to

$$M_m(\bar{z}) = -(u + v)(\bar{z} - \theta_m). \tag{10}$$

This expression is quite general in that $(\bar{z} - \theta_m)$ is simply the distance of the mean phenotype from that expected under mutation equilibrium, and $(u + v)$ is a measure of the mutational restoring force per locus. The essential feature of *Equation (10)* is that mutation acts to reduce the distance between the mean phenotype and $\theta_m$ to a degree that depends on the magnitude of this deviation. *Charlesworth (2013)* implemented a similar mutation model in an investigation of genomic features.

## The stationary distribution of mean phenotypes

Application of *Equations (7) and (10)* to (1b) yields a useful simplification of the stationary distribution that will be adhered to below,

$$\Phi(\bar{z}) = C \cdot \left[\overline{W}(\bar{z})\right]^{2N_e} \cdot \exp\left(\frac{-(\bar{z} - \theta_m)^2}{2\sigma_N^2}\right), \tag{11}$$

with $\sigma_N^2 = \sigma_A^2/[2N_e(u + v)]$. As will be discussed below, under neutrality, the genetic variance $\sigma_A^2$ often scales directly with $N_e$, and population size would have no influence on the distribution in this limiting case, as $\sigma_N^2$ would be independent of $N_e$. More generally, $\sigma_A^2$ is also a function of the intensity of selection, but the bulk of the steady-state distribution will be represented by mean phenotypes that are in the range of effective neutrality with respect to each other, so the scaling relationship of $\sigma_A^2$ under neutrality is expected to be a reasonable first-order approximation.

*Equation (11)* shows that, provided the genetic variance remains roughly constant, the stationary distribution is equal to the product of the expectation under neutrality (where mutation and drift are the only operable evolutionary forces) and the mean fitness function exponentiated by $2N_e$, that is, the stationary distribution is equivalent to a transformation of the neutral expectation by a function of the fitness landscape. Thus, to obtain the overall distribution in the following applications, we require an expression for mean population fitness in terms of the trait mean.

In what follows, insight into the approximate magnitude of $\sigma_N^2$ will be useful. This can be achieved by noting that $2N_e(u + v)$ will have values of the order of magnitude of $4N_e\mu$, where $\mu$ is the mutation rate per nucleotide site. This composite parameter is equivalent to the amount of standing heterozygosity at neutral nucleotide sites in natural populations under mutation-drift equilibrium, and generally ranges from 0.001 to 0.1, with the lower and higher ends of the range being typical in vertebrates and microbes, respectively (*Lynch, 2007*). Thus, because heritabilities $(\sigma_A^2/\sigma_z^2)$ of traits are typically on the order of 0.1 to 0.5 (*Lynch and Walsh, 1998*), $\sigma_N^2$ is expected to be in the range of $1\times$ to $100\times$ the average within-population phenotypic variance for the trait.

### Selection for an intermediate optimum

A commonly assumed form of selection, probably relevant to many cellular features, is the Gaussian (bell-shaped) fitness function with an intermediate optimum phenotype, $\theta_s$, and a width, $\omega$, determining the strength of selection around the optimum,

$$W(z) = \exp\left(-\frac{(z - \theta_s)^2}{2\omega^2}\right). \tag{12}$$

Application of this expression to *Equations (3) and (4)* leads to the expression for mean population fitness, which when applied to *Equation (7)* yields the expression for $M_s(\bar{z})$ necessary for obtaining

**Table 1.** Formulae for mean population fitness, $\overline{W}(\bar{z})$, and the rate of change of the mean phenotype resulting from selection, $M_s(\bar{z})$, obtained from *Equations (4) and (6)*, respectively.

| Model | $\overline{W}(\bar{z})$ | $M_s(\bar{z})$ |
|---|---|---|
| Gaussian | $\sqrt{\dfrac{\omega^2}{\omega^2 + \sigma_z^2}} \cdot \exp\left(-\dfrac{(\bar{z} - \theta_s)^2}{2(\omega^2 + \sigma_z^2)}\right)$ | $-\dfrac{\sigma_A^2(\bar{z} - \theta_s)}{\omega^2 + \sigma_z^2}$ |
| Hyperbolic | $1 - \alpha\, \exp\{-\beta[\bar{z} - (\beta\sigma_z^2/2)]\}$ | $\dfrac{\sigma_A^2 \alpha\beta}{\exp\{\beta[\bar{z} - (\beta\sigma_z^2/2)]\} - \alpha}$ |
| Sigmoid | $\dfrac{1}{1 + \exp[-(\beta/\gamma)(\bar{z} - z^\star)]}$ | $\dfrac{\sigma_A^2(\beta/\gamma)}{1 + \exp[(\beta/\gamma)(\bar{z} - z^\star)]}$ |

DOI: https://doi.org/10.7554/eLife.34820.004

the stationary distribution (*Table 1*). The latter expression shows that the change in the mean phenotype resulting from selection is directly proportional to the deviation of the current mean phenotype from the optimum and inversely proportional to the sum of the squared width of the fitness function and the total phenotypic variance (*Lande, 1976*). As will be seen repeatedly below, phenotypic variance (an inevitable consequence of external environmental and internal cellular effects) generally reduces the efficiency of selection by diminishing the correspondence between genotype and phenotype. If the mean phenotype were to evolve to the optimum, $\bar{z} = \theta_s$, which is highly unlikely with biased mutation pressure, selection would be purely stabilizing in nature, operating only to reduce the variation around the mean.

With both the selection and mutation terms in *Equation (11)* being Gaussian functions, the product is also Gaussian (*Lande, 1976*), in this case leading to a stationary distribution of mean phenotypes

$$\Phi(\bar{z}) = \sqrt{\frac{1}{2\pi\sigma^2(\bar{z})}} \cdot \exp\left(-\frac{[\bar{z} - \mu(\bar{z})]^2}{2\sigma^2(\bar{z})}\right), \tag{13a}$$

with overall mean

$$\mu(\bar{z}) = \frac{(\theta_s/\sigma_S^2) + (\theta_m/\sigma_N^2)}{(1/\sigma_S^2) + (1/\sigma_N^2)} = \frac{\kappa\theta_s + \theta_m}{\kappa + 1}, \tag{13b}$$

and variance

$$\sigma^2(\bar{z}) = \frac{1}{(1/\sigma_S^2) + (1/\sigma_N^2)} = \frac{\sigma_N^2}{\kappa + 1}, \tag{13c}$$

where $\kappa = \sigma_N^2/\sigma_S^2$, with $\sigma_S^2 = (\omega^2 + \sigma_z^2)/(2N_e)$ and $\sigma_N^2$ (as defined as above) being the variances of the contributions associated with selection and mutation.

*Equation (13b)* states that the grand mean is equal to a weighted average of the expectations under mutation and selection alone (each component being weighted by the inverse of the variance of the function). *Equation (13c)* states that the variance of means is equal to half the harmonic mean of the variances associated with selection and mutation alone. As $\sigma_S^2 \to \infty$, which implies a flatter fitness function and hence an approach toward neutrality, the mean and variance converge on the expectations for a purely mutationally driven process, $\theta_m$ and $\sigma_N^2$. As $\sigma_N^2 \to \infty$, which implies a weakening influence of mutation on the overall distribution, the mean and variance converge on the expectations for a purely selection-driven process, $\theta_s$ and $\sigma_S^2$.

As can be seen from *Equations (13b, c)*, a key determinant of the form of the stationary distribution of means is the composite parameter $\kappa = \sigma_A^2/[2(u + v)(\omega^2 + \sigma_z^2)]$, which the following observations suggest is generally $\gg 1$. First, the width of the fitness function $\omega$ can be expected to be generally greater than the phenotypic standard deviation $\sigma_z$, else the selective load on the trait would be enormous, and this is indeed generally observed (*Walsh and Lynch, 2018*). Given the range of heritability estimates noted above, this implies that the ratio $\sigma_A^2/(\omega^2 + \sigma_z^2)$ is unlikely to be

greater than 0.1 under strong selection, and can become one to two orders of magnitude smaller than 0.1 under weak selection. Second, mutation rates at the single nucleotide level are typically in the range of $10^{-11}$ to $10^{-8}$, with the former being approached in microbes and the latter in large multicellular species (*Lynch et al., 2016*). Thus, keeping in mind that individual targets of mutation may comprise more than single nucleotide sites, $1/[2(u + v)]$ is still likely to be in the range of $10^7$ to $10^{10}$. Together, these results suggest a likely range for $\kappa$ of $10^4$ to $10^9$, which simplifies *Equations (13b, c)* to

$$\mu(\bar{z}) \simeq \theta_s + (\theta_m/\kappa) \tag{14a}$$

$$\sigma^2(\bar{z}) \simeq \sigma_N^2/\kappa = \sigma_S^2. \tag{14b}$$

With these parameter values in mind, *Figure 2* shows that the form of the stationary distribution varies dramatically with the value of $\sigma_N^2/\kappa = \sigma_S^2$, becoming extremely narrow and extremely flat at opposite ends of the spectrum for this key composite parameter. The degree to which $\theta_m$ deviates from $\theta_s$ for cellular features is unknown, but there is no reason to expect them to be equal. If they differ greatly, $\mu(\bar{z})$ can substantially deviate from the optimum to a degree that depends on the weighting factor $\kappa$ (*Figure 2*).

## Hyperbolic fitness function

Many cellular features are likely to be primarily under continuous selection for an extreme optimum, but with diminishing strength of selection as the optimum is approached. For example, many

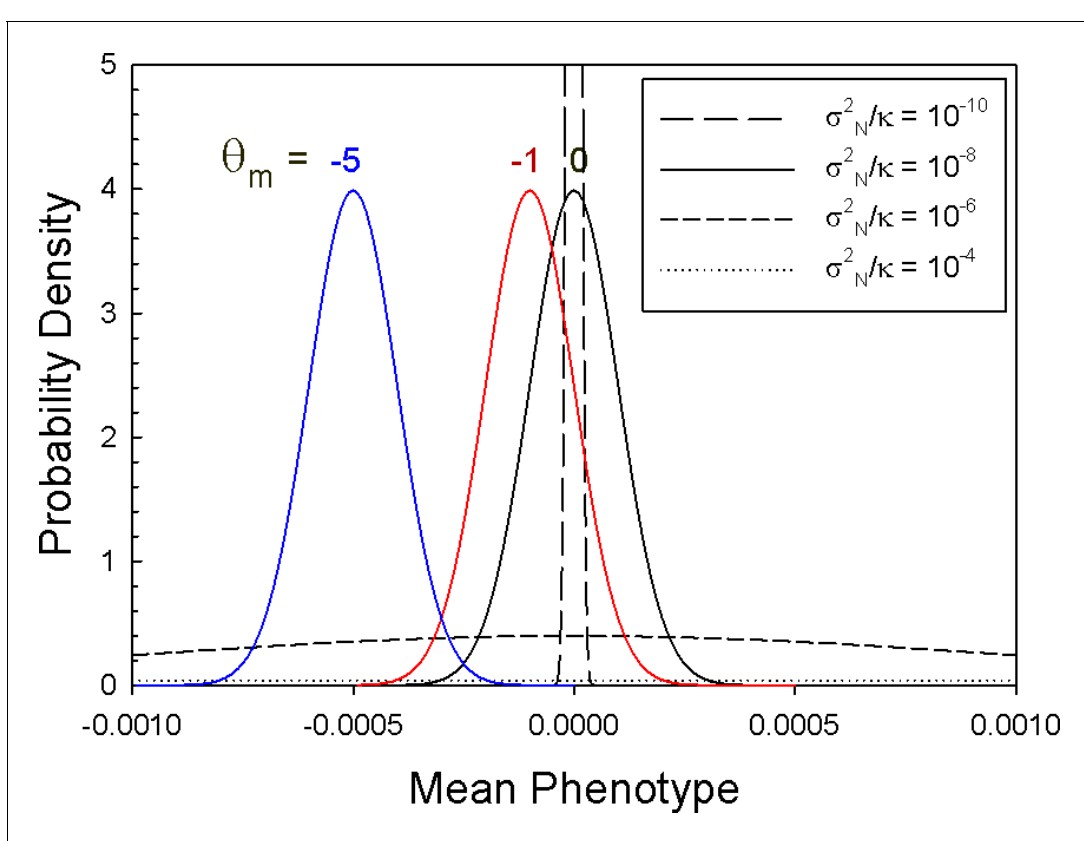

**Figure 2.** Stationary distributions of mean phenotypes, with optimum phenotype $\theta_s = 0$ and $\kappa = 10^4$. Results are given for three different values of $\theta_m$ for the condition in which $\sigma_N^2/\kappa = 10^{-8}$ (colored curves), and four different values of $\sigma_S^2$ for the case in which the mutational mean coincides with the optimum (black curves). The parametric values used in these plots assume a scale on which the phenotypic standard deviation is 1.0, so a mean phenotype of ±0.001 is equivalent to a shift of 0.1% phenotypic standard deviations from the optimum.

DOI: https://doi.org/10.7554/eLife.34820.005

enzymes are likely to be selected for as high a catalytic rate as possible, protein structures for as high folding rates and stability as possible, binding interfaces with as high affinities as possible, etc. One way of representing this type of selection involves the hyperbolic function,

$$W(z) = 1 - \alpha \exp(-\beta z), \tag{15}$$

where the constants $0 \leq \alpha \leq 1$ and $\beta \geq 0$, respectively, define the amplitude and rapidity of the fitness response to increasing $z$. Fitness is equal to $1 - \alpha$ when $z = 0$, and asymptotically approaches one as $z \to \infty$.

Expressions for the mean population fitness and the change in the mean resulting from selection, obtained by the procedures noted above, are provided in *Table 1*, and substitution of the former into *Equation (11)* yields the stationary distribution of mean phenotypes. Because of the asymmetry of this fitness function, the resultant distribution is no longer perfectly Gaussian, but setting $\partial \Phi(\bar{z})/\partial \bar{z} = 0$ yields an expression for the single mode of the distribution, $z$

$$(\hat{z} - \theta_m)\{\exp[\beta \hat{z}] - \phi\} = 2N_e \beta \sigma_N^2 \phi, \tag{16a}$$

with $\phi = \alpha \exp(\beta^2 \sigma_{\bar{z}}^2 / 2)$. Despite the monotonic increase in fitness with $z$, the distribution of mean phenotypes is prevented from progressive increase by the counteraction of mutation and the diffusive action of drift. Because selection is always in the positive direction, the expected mode always exceeds the neutral expectation $\theta_m$, to a degree that increases with the effective population size. *Equation (16a)* is readily solved numerically, but provided $\beta \hat{z} < 1$, in the limit of large $N_e$,

$$\hat{z} \simeq \theta_m + \sqrt{2N_e \phi \sigma_N^2}. \tag{16b}$$

Although the hyperbolic fitness function generates a slightly asymmetric distribution of means (with tail to the right), the bulk of the distribution is approximately normal, and an excellent approximation to the variance can be obtained from the curvature of the stationary distribution around the mode (using the negative of the inverse of the second derivative of the stationary distribution),

$$\sigma^2(\bar{z}) \simeq \frac{1}{\left[2N_e \beta^2 \phi \exp(\beta \hat{z})/(\exp(\beta \hat{z}) - \phi)^2\right] + (1/\sigma_N^2)} \tag{17}$$

As in the case of the Gaussian fitness function, *Equation (13c)*, the two terms in the denominator are respectively the inverses of the variances expected under the limits of strong selection and neutrality.

An example of the influence of population size on the stationary distribution is given in *Figure 3*, where there is a strong mutational bias away from the optimum. The distributions progressively move to the right with an increase in $N_e$, with the mean phenotype increasing five-fold over a three order-of-magnitude range of $N_e$. As can be seen from *Equation (16b)*, equal changes in either $N_e$ or the neutral variance $\sigma_N^2$ have identical effects on the mean, although effects on the variance are opposite in direction.

## Sigmoid fitness function

Finally, we consider a variant of the fitness function just noted. With the previous fitness function, *Equation (15)*, the selection gradient progressively declines with increasing phenotypic value over the full range of $z$, with increasing $z$ resulting in an asymptotic approach to maximum fitness. With a sigmoid fitness function, sometimes called a mesa function (*Gerland and Hwa, 2002*; *Berg et al., 2004*), there is an inflection point such that the fitness landscape becomes progressively flatter at both higher and lower values. This means that adjacent variants become increasingly similar in fitness (i.e. more neutral with respect to each other) at both extremes of the phenotype distribution.

The sigmoid fitness function for individual phenotypes can be described as

$$W(z) = \frac{1}{1 + \exp[-\beta(z - z^\star)]}, \tag{18a}$$

where $z^\star$ denotes the inflection point at which $W(z) = 0.5$. This function is closely approximated by

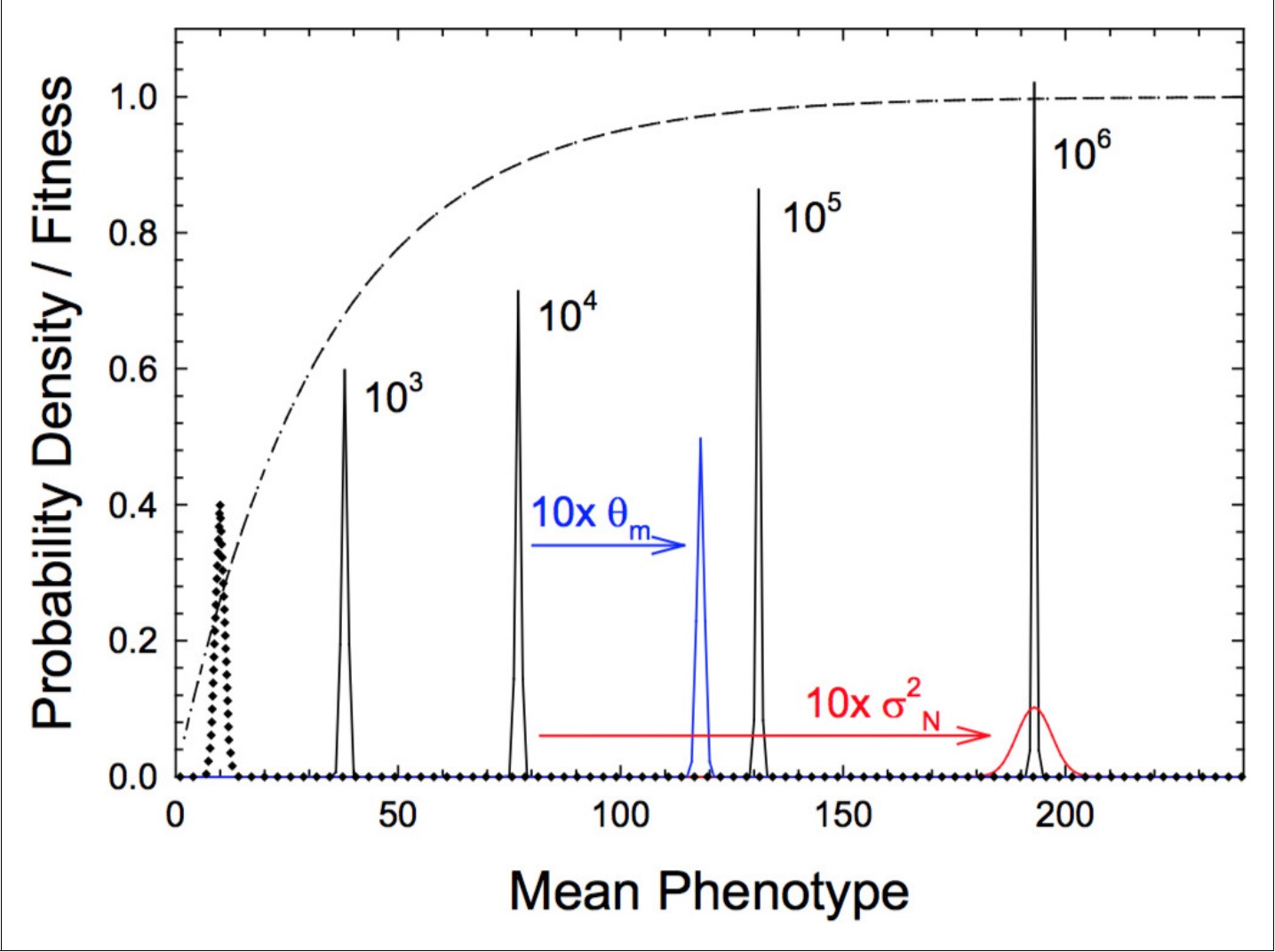

**Figure 3.** Stationary distributions of mean phenotypes with a hyperbolic fitness function, Equation (14) with $\alpha = 1$ and $\beta = 0.03$ and denoted by the dashed line. Black distributions denote results for various effective population sizes ($N_e = 10^3$ to $10^6$), with phenotypic variance $\sigma_z^2 = 1$, mutational variance $\sigma_N^2 = 1$, and the mean phenotype under neutrality (distribution given by the dotted line) $\theta_m = 10$. Blue and red distributions denote the effects of altering $\theta_m$ and $\sigma_N^2$.

DOI: https://doi.org/10.7554/eLife.34820.006

$$W(z) = \frac{1}{2}\left[1 + \mathrm{erf}\left(\frac{\sqrt{\pi}\beta(z - z^*)}{4}\right)\right],\qquad(18b)$$

where erf is the error function (the cumulative standard normal distribution), which facilitates integration with *Equation (4)*. The resultant expression for mean population fitness is also sigmoid, but with phenotypic variance reducing the strength of selection from $\beta$ to $\beta/\gamma$, where $\gamma = \sqrt{1 + (\beta^2 \sigma_z^2 \pi/8)}$ (*Table 1*).

As in the case of the hyperbolic fitness function, the mesa function does not yield a perfectly Gaussian distribution of mean phenotypes, but an expression for the mode ($z$) can be acquired using the methods noted above,

$$(\hat{z} - \theta_m)\{\exp[(\beta/\gamma)(\hat{z} - z^*)] + 1\} = 2N_e(\beta/\gamma)\sigma_N^2,\qquad(19a)$$

which again has a single solution, indicating a unimodal stationary distribution. If $\beta\hat{z}/\gamma<1$, in the limit of large $N_e$,

$$\hat{z} \simeq \theta_m + \sqrt{2N_e\sigma_N^2\exp(\beta z^\star/\gamma)}, \tag{19b}$$

which has a form similar to the expression noted with the hyperbolic fitness function. From the form of these equations, it can again be seen that there are several equivalent effects of the underlying parameters. For example, a doubling of $N_e$ has the same effect as a doubling of $\sigma_N^2$ on the mode, and a doubling of $\beta$ the same effect as a reduction in $\gamma$ by 50%. Although more complicated, the expected variance in means under the sigmoid model is similar in form to that noted above for the Gaussian and hyperbolic fitness functions,

$$\sigma^2(\bar{z}) \simeq \frac{1}{\left\{2N_e(\beta/\gamma)^2\exp[-(\beta/\gamma)(\bar{z}-z^\star)]/(1+\exp[-(\beta/\gamma)(\bar{z}-z^\star)])^2\right\} + (1/\sigma_N^2)} \tag{20}$$

## Discussion

The preceding models are meant to provide heuristic guidance into the evolutionary mechanisms responsible for the dispersion of mean phenotypes of a diversity of subcellular and molecular features. Although such traits may sometimes be under selection for an intermediate optimum, selection may often operate in a continuous directional fashion. In either case, there are two reasons why mean phenotypes are unlikely to commonly achieve states that endow a population with maximum fitness. First, if mutation bias conflicts with the directional effects of selection, the optimum phenotype will not coincide with the mean phenotype. Second, even in the absence of mutation bias and regardless of the form of the fitness function, a drift barrier exists beyond which the gradient of the selection function is not steep enough to overcome the vagaries of genetic drift, thereby preventing further adaptive progress. Within the confines of the drift barrier, the mean phenotype will wander to a degree that depends on the strength of local patterns of mutation and selection.

These points have implications for the degree to which the 'adaptive paradigm' should be embraced as an explanatory framework for diversification at the cellular level. For example, with mutation bias encouraging the mean phenotype to deviate from the optimum, the result will be a population under persistent directional selection despite the existence of an attainable (but not sustainable) phenotype with maximum fitness. Even without mutation pressure and in the face of intrinsic directional selection, for example, a hyperbolic or mesa fitness function, the most common mean phenotype will not be equivalent to the optimum phenotype, and the drift barrier will ensure variation in mean phenotypes among populations exposed to identical selection pressures.

An attempt has been made to couch the stationary forms of mean-phenotype distributions in terms of underlying parameters that are at least in principle observable empirically. Consider, for example, the model for stabilizing selection for a specific optimum. From *Equation (14a)*, the expected deviation of the mean phenotype from the optimum resulting from mutation bias is $\theta_m/\kappa$, which expands to $\theta_m\left[2(u+v)(\omega^2+\sigma_z^2)/\sigma_A^2\right]$, a somewhat complex function that may not be immediately transparent. However, a wide variety of models suggest that $\sigma_A^2$ scales directly with $N_e\mu$ provided selection is weak (*Bürger et al., 1989*; *Zeng and Cockerham, 1993*; *Charlesworth, 2013*), and because $u$ and $v$ (the forward and reverse mutation rates) are both proportional to $\mu$ (the total mutation rate per site), this implies that the average deviation of the mean from the optimum scales as $\theta_m(\omega^2+\sigma_z^2)/N_e$, or approximately as $\theta_m\omega^2/N_e$ assuming weak selection. Thus, the deviations of phenotypic means from the selective optimum are expected to be inversely proportional to $N_e$, a point also made by *Charlesworth (2013)* in a somewhat different analysis. Note, however, that this is only the expected pattern, as the mean phenotype is still expected to drift above and below the expectation to a degree depending on the effective strength of selection. As noted in *Equation (14b)*, and previously pointed out by *Lande (1975)* and *Lande (1976)*, the magnitude of this drift variance is also inversely proportional to $N_e$, which implies that the standard deviation with respect to the expected mean scales as $\sim 1/\sqrt{N_e}$.

Of course, $\theta_m$ (the mean phenotype expected under neutrality) may differ among lineages and the within-population genetic variance $\sigma_A^2$ is sensitive to the strength of selection, in which case the power to detect such relationships may be challenging. In addition, the linear scaling of $\sigma_A^2$ with $N_e$ is

unlikely to continue indefinitely, unless $N_e$ in natural populations rarely attains levels where all constituent loci are saturated with segregating mutations. The salient issue is that the preceding expressions provide qualitative insight into the behavior of mean phenotypes in alternative population-genetic environments, while also revealing the types of measurements that need to be made if we are to understand such behavior. For example, we know essentially nothing about the key mutational ($\theta_m$) and selection ($\omega^2$) parameters for cell biological features and how these might vary among species. This is not a trivial issue, as the influence of both parameters in determining the most likely locations of mean phenotypes are just as central as the role played by $N_e$.

Applying the same logic to results for plateaued fitness functions leads to the prediction that the expected mode of mean phenotypes will scale fairly strongly with the effective population size, in the limit approaching proportionality to $\sqrt{N_e}$, that is, a 10-fold increase in the mean phenotype with a 100-fold increase in $N_e$. As shown in **Figure 3**, a simple change in the mutational variance $\sigma_M^2$ (with no associated change in mutational bias) can also cause a substantial shift in the position of the mean phenotype. These sorts of observations raise the significant possibility that species with substantially different population-genetic environments may commonly exhibit measurable differences in trait means despite experiencing identical forms of directional selection, again raising challenging issues for those who wish to interpret phenotypic differences as reflections of different underlying processes of selection.

Although the data are not extensive, several lines of evidence support the idea that the mean phenotypes of cellular attributes are indeed modulated by the power of random genetic drift. The most compelling example derives from observations on the mutation rate (per nucleotide site per generation), which scales approximately inversely with the 1000-fold range of variation in $N_e$ across the Tree of Life (**Lynch et al., 2016**). Such a scaling is qualitatively consistent with the drift-barrier hypothesis for mutation-rate evolution (**Lynch, 2010**; **Lynch, 2011**), which postulates that because most mutations are deleterious, selection will typically operate to improve replication fidelity, with refinements in molecular performance eventually being thwarted by random genetic drift – as the mutation rate is progressively lowered, there is less room for improvement and hence a narrower range of selectively advantageous replication-fidelity variants accessible by selection.

Enzyme efficiency provides a second broad category of traits with evolutionary behavior seemingly in accordance with the theory outline above. For example, **Bar-Even et al. (2011)** have found that enzymes involved in secondary metabolism are on average $\sim 30 \times$ less efficient than those involved in central metabolism, suggesting that selection operates less effectively on enzymes further removed from core energetic determinants. More directly relevant to the points made above, **Bar-Even et al. (2011)** also found that prokaryotic enzymes have slightly better kinetics than those from eukaryotes, as expected for species with higher effective population sizes and consistent with the prediction that improvement of enzyme efficiencies will stall once the gradient of the fitness surface is on the order of $1/N_e$ (**Hartl et al., 1985**). The fact that bacteria utilize transcription-factor binding-site motifs with stronger affinity to their cognate transcription factors than is the case in eukaryotes is also plausibly related to a higher efficiency of selection in the former (**Lynch and Hagner, 2015**).

Finally, proteins typically evolve to the 'margin of stability,' such that only one or two mutations are usually enough to destabilize the folding process (**Taverna and Goldstein, 2002**; **Tokuriki and Tawfik, 2009**). Protein stability is deemed to be positively associated with fitness because destabilized proteins are prone to loss of function, aggregation, and/or direct toxicity. Strikingly, however, it is relatively easy to obtain more stable proteins by mutagenesis (**Matsuura et al., 1999**; **Bershtein et al., 2013**; **Sullivan et al., 2012**), with the contributing residues typically interacting in an additive fashion (**Wells, 1990**; **Serrano et al., 1993**; **Zhang et al., 1995**). Moreover, although it is commonly argued that marginal stability is required for proper protein function, with excess stability somehow reducing protein performance, this has not held up to close scrutiny. Many examples exist in which increased stability has been achieved in laboratory modifications of proteins with few if any consequences for enzyme efficiency (e.g. **Giver et al., 1998**; **Zhang et al., 1995**; **Taverna and Goldstein, 2002**; **Borgo and Havranek, 2012**; **Moon et al., 2014**).

These observations suggest that despite persistent selection for high folding stability, the plateau-like nature of the fitness landscape results in diminishing fitness advantages of increasing stability. A hyperbolic relationship between fitness and the binding energy involving protein stability

follows from biophysical principles (*Govindarajan and Goldstein, 1997*; *Taverna and Goldstein, 2002*; *Bloom et al., 2005*; *Zeldovich and Shakhnovich, 2008*; *Wylie and Shakhnovich, 2011*; *Serohijos and Shakhnovich, 2014*), and under this model, proteins are expected to be pushed by natural selection to more stable configurations until reaching the point where any further fitness improvement is small enough to be offset by the vagaries of random genetic drift and/or mutation pressure towards less stable states. Notably, proteins of equivalent length fold at least ten times more rapidly in bacteria than in eukaryotes (*Galzitskaya et al., 2011*). Moreover, an in vitro evaluation of the folding stability of the dihydrofolate reductase enzyme from 36 species of mesophilic bacteria illustrates the existence of a substantial range of variation among species, with the standard deviation being roughly 10% of the mean (*Bershtein et al., 2015*). In principle, such a distribution may reflect the dispersion in mean phenotypes associated with drift around the drift barrier.

Although the mutation function employed here likely comes closer to approximating the situation for cellular features than do previous functions relied on in quantitative genetics, in reality we do not know the exact form of this function for any cellular feature. Thus, the mathematical theory developed here is best viewed as a guide to approaching the problem at hand rather than as an indelible platform for quantitative analysis. Despite such uncertainties, however, the central feature of the theory presented above is that, regardless of the form of the underlying mutation and selection functions, the stationary distribution of mean phenotypes can generally be viewed as the product of the pattern expected under neutrality alone and the associated function for mean population fitness taken to the $2N_e$ power, as described by *Equations (1a,b) and (11)*. Similar behavior was previously pointed out for the stationary distribution of allele frequencies (*Wright, 1969*). Thus, once the key underlying functions have been elucidated, the precise details of the theory can be readily modified with alternative mathematical functions.

Finally, a key issue that is not formally evaluated here, but is arguably relevant to a number of cellular features, concerns the matter of peak shifts across the stationary distribution. Questions regarding this matter are typically inspired by Wright's (1932) metaphor of an adaptive topography, with multiple fitness peaks and valleys of various depths over the phenotypic landscape. However, unless the distribution of mutational effects is completely flat, the relevant topography is not simply defined by the fitness landscape but by the joint action of both selection and mutation. Although the stationary distribution was unimodal in all of the cases examined above, plausible cases exist in which the stationary distribution exhibits two peaks, one largely driven by selection and the other by mutation pressure. For this to occur, the gradient of mutation pressure in one direction has to be of a form such that its product with the selection gradient has an internal minimum (*Figure 4*). In principle, this can happen when at the intersection of intermediate phenotypes the two functions are sufficiently upwardly concave that their product reaches a local minimum.

Under such a scenario, the population mean phenotype is expected to reside in two alternative semi-stable domains for extended periods of time, with the rates of transitions between domains depending on the relative heights of the two peaks, the depth of the distributional valley, and the curvatures of the stationary distribution at the inflection points (*Lande, 1985*; *Barton and Rouhani, 1987*). Over long evolutionary time periods, such a system will exhibit detailed balance – the net fluxes will be equal in both directions, with the ratio of the occupancy of the two alternative domains being inversely related to the ratio of the transition rates between them, that is, with the less frequent domain having a higher conditional rate of transition to the more frequent domain.

Although the frequency of stationary distributions with multimodal forms is unknown, they have been predicted to arise in some situations involving transcription (*Lynch and Hagner, 2015*; *Tuğrul et al., 2015*). Should they exist, the picture from comparative analyses would be one of qualitative changes in mean phenotypes in adjacent lineages. Tempting as it might be to invoke shifting ecological pressures to explain such patterns, they would be occurring in the absence of any underlying changes in selection, being a simple consequence of the multiplicity of mutational opportunities in one direction balanced by selective pressures in the other. Such ideas may be helpful in attempts to decipher the substantial and seemingly disorganized diversity of certain cellular features such as open vs. closed mitosis (*Sazer et al., 2014*), the structure of the centrosome (*Carvalho-Santos et al., 2011*), and the variable multimeric states of proteins (*Dayhoff et al., 2010*; *Lynch, 2013*; *Ahnert et al., 2015*) across the Tree of Life.

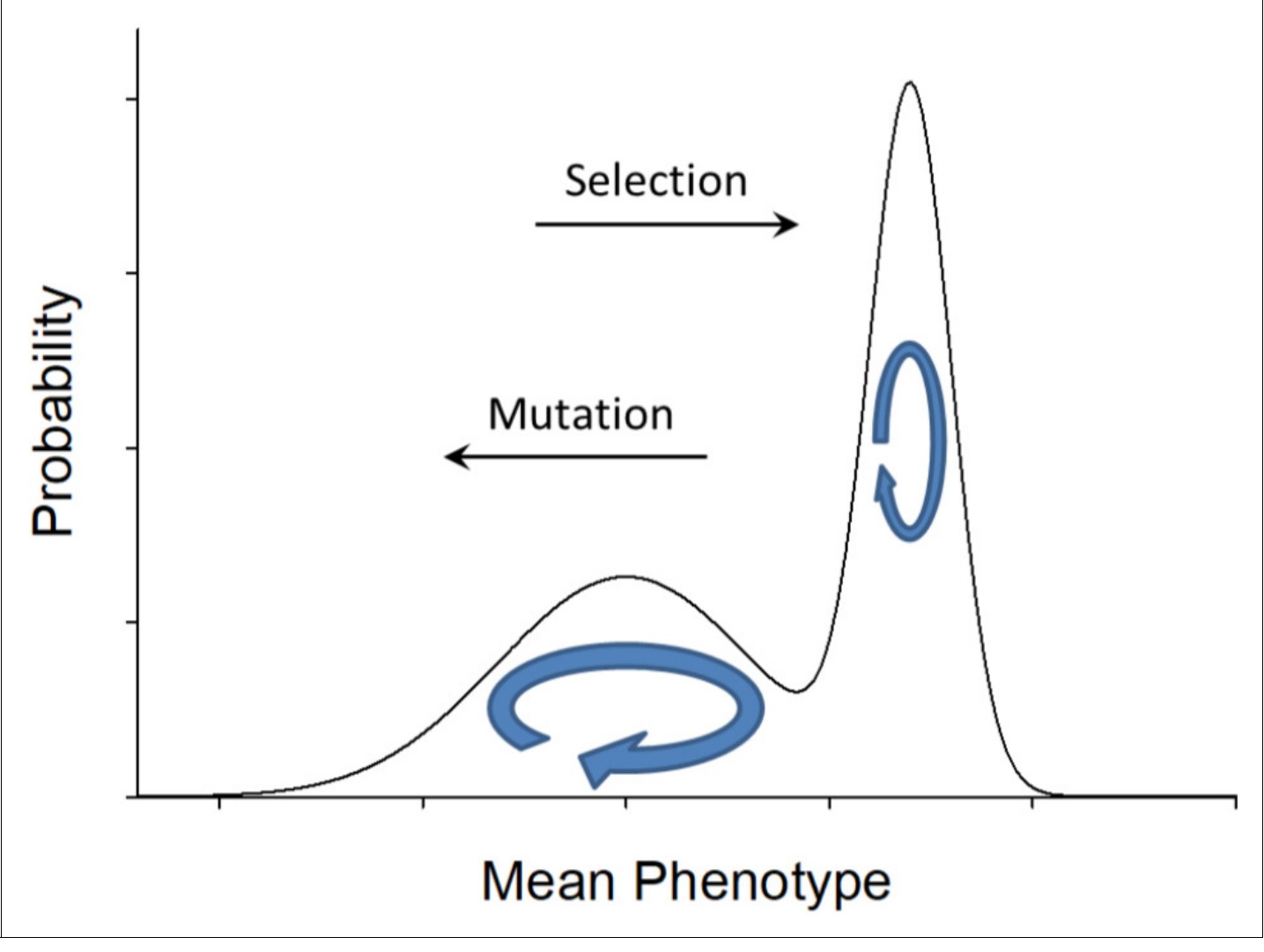

**Figure 4.** An example of a bimodal stationary distribution of mean phenotypes, with mutation pressure to the left and selection pressure to the right, with the forms of these two functions being such that their product is a minimum at the valley in the landscape. Most of the time, a population will reside in one domain or the other, wandering over a range of phenotype space to the left or the right of the valley, but occasionally a transition will be made across the valley impelled by a stochastic series of drift and mutational events. Populations crossing the valley to the right are pulled by selection pressure, whereas those crossing to the left are pulled by the multiplicity of mutational opportunities.
DOI: https://doi.org/10.7554/eLife.34820.007

## Acknowledgements
I thank M Bauer, J Felsenstein, P Higgs, P Johri, M Lässig, M Manhart, A Moses, and D Needleman for helpful comments. This research was supported in part by the National Science Foundation under Grant No. PHY11-25915 to the Kavli Institute of Theoretical Physics. Support was also provided by the Multidisciplinary University Research Initiative awards W911NF-09-1-0411 and W911NF-09-1-0444 from the US Army Research Office, National Institutes of Health awards R01-GM036827 and R35-GM122566-01, and National Science Foundation award MCB-1518060.

## Additional information

### Funding

| Funder | Grant reference number | Author |
|---|---|---|
| Army Research Office | W911NF-09-1-0444 | Michael Lynch |
| Army Research Office | W911NF-14-1-0411 | Michael Lynch |
| National Institutes of Health | R01-GM036827 | Michael Lynch |
| National Institutes of Health | R35-GM122566 | Michael Lynch |
| National Science Foundation | PHY11-25915 | Michael Lynch |

The funders had no role in study design, data collection and interpretation, or the decision to submit the work for publication.

### Author contributions

Michael Lynch, Formal analysis, Funding acquisition, Investigation, Methodology, Writing—original draft, Project administration, Writing—review and editing

### Author ORCIDs

Michael Lynch (iD) http://orcid.org/0000-0002-1653-0642

### Decision letter and Author response

Decision letter https://doi.org/10.7554/eLife.34820.010
Author response https://doi.org/10.7554/eLife.34820.011

## Additional files

### Supplementary files

• Transparent reporting form
DOI: https://doi.org/10.7554/eLife.34820.008

This is a theory paper, hence it contains no data.

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
