## [Decision Letter]

Thank you for submitting your article "Phylogenetic Divergence of Cell Biological Features" for consideration by *eLife*. Your article has been reviewed by two peer reviewers, and the evaluation has been overseen Naama Barkai as the Senior and Reviewing Editor. The following individual involved in review of your submission has agreed to reveal his identity: Joe Felsenstein (Reviewer #2).

The reviewers have discussed the reviews with one another and the Reviewing Editor has drafted this decision to help you prepare a revised submission.

Please address all comments below most of which can be addressed with suitable changes in the text. Others require a response which the reviewers will consider in rendering a final determination.

*Reviewer #1:*

I reviewed this manuscript with a PhD student from my laboratory.

Whether cellular traits are at the optimal configuration allowed by biophysical constraints in any organism is unknown. As was shown in population genetics, genetic drift could play an important role in preventing traits from reaching this optimum. However, biased mutation pressures could also contribute to drive average phenotypes away from their optimum. The extent of this effect is unknown. This paper addresses the evolution of cellular traits, particularly the relative contribution of natural selection, mutation and genetic drift on their evolution and divergence among populations or species.

Lynch develops a theoretical framework to identify the conditions in which phenotypes can be driven away from optima by biased mutational pressure. He also introduces different fitness functions that likely apply to cellular traits and shows how these fitness functions interact with various mutational effects to affect the expected stationary trait distribution.

The paper is well written and the findings are important because, among other things, they stimulate new avenues of investigation in cell biology. The presentation however could limit the accessibility of the manuscript to the broad audience of *eLife*.

It would be useful in the Introduction to mention why cellular traits require a special treatment and approach, and why they cannot just be considered as other quantitative traits. The author mentions that since they evolve in a relatively stable environment due to homeostasis, cellular traits could be under relatively uniform selective pressure, even among distinct lineages. However, wouldn't that be true for internal organs also in multicellular species? Also, homeostasis is the result of these cellular traits interacting with the environment, so it cannot be considered as an independent factor. Maybe cellular traits are just a special case of slowly evolving traits?

The first paragraph of the Introduction appears to be reporting observations that are commonly known for evolutionary biologists. However, *eLife* not being specialized in evolution, it could be useful to cite some general references for these statements. The second paragraph of the subsection “The Process of Mutation” could also be supported by more references, same thing for the Discussion. Along the same line, the readership of *eLife* would appreciate some graphical and simplified representations of the processes described with equations, some sort of graphical summary of the paper.

The Introduction mentions examples of cellular traits. It could be useful to mention them earlier to introduce what cellular traits are.

The paragraph above Equation 1B: does this partition of the variance of z implicitly assumes that the parental average (A) does not covary with the deviation from additivity (e)? It would be clearer to briefly state this assumption. Some stochastic effects in cell biology could be noise factors that are correlated positively or negatively with the average effects.

Finally, I do not have the skills required to fully verify the validity of the different derivations and mathematical assumptions presented. I assume another reviewer will have done so.

*Reviewer #2:*

This is a very interesting and important paper. Where a lot of us have been deterred from thinking hard about this by the worry that characters at the molecular level might involve too few genes to be successfully modeled by the machinery of evolutionary quantitative genetics, Lynch has plunged in and obtained some very interesting results.

I have several substantive questions, and then a bunch of suggestions for clarity. 1) “distributions expected under selection along and under mutation alone” are invoked. The latter I can see. But the former is not obvious. I get the impression from the equations that this is the distribution that would result under selection-versus-drift when the additive genetic variance is held constant at σ 2 A. So there is some kind of assumption that, in the case of “selection alone”, selection does not erode the additive genetic variance.

2) “evolution by natural selection comes to a standstill where.… the phenotypic mean resides at the point where the slope of the fitness function is zero.” I wonder whether that is true. I recall that if the fitness function (curve of fitness as a function of phenotype) is a mixture of two Gaussian peaks, one smaller than the other, where the phenotypic mean will come to a standstill depends on the slope, not of that fitness curve, but of the fitness curve where each component is fattened by the additive genetic variance. That can be quite different. Or do I misunderstand what is being said here?

3) The mathematics here uses compound parameters which determine the final equilibrium distribution. However should it not be noted that these do determine the time dynamics in a population. If one is looking at a phylogeny of species, rather than species that have diverged long enough that each is in its equilibrium distribution, one does need the parameters that determine the time dynamics. This should at least be mentioned, as multispecies data tend to come to us at the tips of phylogenies.

---

## [Author Response]

Reviewer #1:

[…] It would be useful in the Introduction to mention why cellular traits require a special treatment and approach, and why they cannot just be considered as other quantitative traits. The author mentions that since they evolve in a relatively stable environment due to homeostasis, cellular traits could be under relatively uniform selective pressure, even among distinct lineages. However, wouldn't that be true for internal organs also in multicellular species? Also, homeostasis is the result of these cellular traits interacting with the environment, so it cannot be considered as an independent factor. Maybe cellular traits are just a special case of slowly evolving traits?

I have attempted to reword the Introduction a bit, and elaborate elsewhere, to accommodate the reviewer’s points here. I do not mean to imply that cellular traits need to be treated differently than other classical quantitative traits, but rather that due to relative homeostasis (and the likely more constant selective environment), they may actually fulfill the assumptions of the models better. The point about internal organs is interesting, and has now been mentioned.

The first paragraph of the Introduction appears to be reporting observations that are commonly known for evolutionary biologists. However, eLife not being specialized in evolution, it could be useful to cite some general references for these statements. The second paragraph of the subsection “The Process of Mutation” could also be supported by more references, same thing for the Discussion. Along the same line, the readership of eLife would appreciate some graphical and simplified representations of the processes described with equations, some sort of graphical summary of the paper.

As requested, additional references are cited – the Lynch and Walsh / Walsh and Lynch books give a very broad and up-to-date overview of theoretical and empirical basis of quantitative genetics. I have also attempted to produce a figure that I hope will soften things a bit and provide additional clarity.

The Introduction mentions examples of cellular traits. It could be useful to mention them earlier to introduce what cellular traits are.

This is now mentioned explicitly in the first paragraph of the paper.

The paragraph above Equation 1B: does this partition of the variance of z implicitly assumes that the parental average (A) does not covary with the deviation from additivity (e)? It would be clearer to briefly state this assumption. Some stochastic effects in cell biology could be noise factors that are correlated positively or negatively with the average effects.

This is now stated explicitly at the designated location.

Finally, I do not have the skills required to fully verify the validity of the different derivations and mathematical assumptions presented. I assume another reviewer will have done so.

Reviewer #2:

[…] I have several substantive questions, and then a bunch of suggestions for clarity. 1) “distributions expected under selection along and under mutation alone” are invoked. The latter I can see. But the former is not obvious. I get the impression from the equations that this is the distribution that would result under selection-versus-drift when the additive genetic variance is held constant at σ 2 A. So there is some kind of assumption that, in the case of “selection alone”, selection does not erode the additive genetic variance.

Yes, I believe that is the correct interpretation, and this is now stated explicitly and elaborated upon more in the Discussion, with further justification noted in the section on the stationary distribution.

2) “evolution by natural selection comes to a standstill where.… the phenotypic mean resides at the point where the slope of the fitness function is zero.” I wonder whether that is true. I recall that if the fitness function (curve of fitness as a function of phenotype) is a mixture of two Gaussian peaks, one smaller than the other, where the phenotypic mean will come to a standstill depends on the slope, not of that fitness curve, but of the fitness curve where each component is fattened by the additive genetic variance. That can be quite different. Or do I misunderstand what is being said here?

I have reworded things to make clear that the formulation is referring to mean population fitness with respect to the mean phenotype, under the stated assumption that the phenotype distribution is normal.

3) The mathematics here uses compound parameters which determine the final equilibrium distribution. However should it not be noted that these do determine the time dynamics in a population. If one is looking at a phylogeny of species, rather than species that have diverged long enough that each is in its equilibrium distribution, one does need the parameters that determine the time dynamics. This should at least be mentioned, as multispecies data tend to come to us at the tips of phylogenies.

Yes, this is an important point, and at the beginning of the section on the model, I now state “From an empirical perspective, this steady-state view of evolution implicitly assumes that enough time has elapsed between observed taxa that the dynamics of the evolutionary process are of negligible significance (which would not be the case for closely related species).